# Gold Nanoparticles Green-Synthesized by the *Suaeda japonica* Leaf Extract and Screening of Anti-Inflammatory Activities on RAW 267.4 Macrophages

Gi-Young Kwak [1,†], Yaxi Han [1,†], Sul Baik [1], Byoung-Man Kong [1], Deok-Chun Yang [1], Se-Chan Kang [1,*] and Johan Sukweenadhi [2,*]

1   Department of Oriental Medicinal Biotechnology, College of Life Sciences, Kyung Hee University, Yongin-si 17104, Gyeonggi-do, Korea; kwakgiyoung8@gmail.com (G.-Y.K.); navycki@gmail.com (Y.H.); bssul_w@nate.com (S.B.); kong2167@naver.com (B.-M.K.); dcyang@khu.ac.kr (D.-C.Y.)
2   Department of Plant Biotechnology, Faculty of Biotechnology, University of Surabaya, Kalirungkut, Surabaya 60293, Indonesia
*   Correspondence: sckang@khu.ac.kr (S.-C.K.); sukwee@staff.ubaya.ac.id (J.S.)
†   These authors contributed equally to this work.

**Abstract:** Biosynthesis of gold nanoparticles from medicinal plants has become a modern strategy in biomedical research based on their exclusive properties, including specific targeting, lower toxicity, and biocompatibility. In this study, gold nanoparticles, reduced by the *Suaeda japonica* leaf extract, were promptly validated by UV–visible (UV–Vis) spectroscopy at 548 nm. No additional reducing agents were needed in this kind of a reduction reaction, which provided evidence of green synthesis. Dynamic light scattering (DLS), energy-dispersive X-ray spectroscopy (EDX), field-emission transmission electron microscopy (FE-TEM), selected area electron diffraction (SAED), and X-ray diffraction (XRD) analyses were used to illustrate the nanoscale characterization of *S. japonica* gold nanoparticles (Sj-AuNps). Furthermore, the cytotoxicity effect of Sj-AuNps against the RAW 264.7 cell line was determined by performing an MTT assay. We also investigated Sj-AuNps' anti-inflammatory properties in LPS-induced murine macrophages. These nanoparticles reduced the generation of nitric oxide (NO) and prostaglandin E2 ($PGE_2$) and repressed the expression of the LPS-stimulated inducible nitric oxide synthase (*iNOS*) and cyclooxygenase-2 (*COX-2*) genes. This study presents a significant biomedical application of *S. japonica* AuNps. The anti-inflammatory capabilities of Sj-AuNps underline their potential as possible options for suppressing inflammation-mediated diseases.

**Keywords:** anti-inflammatory; biosynthesis; gold nanoparticles; *Suaeda japonica*





## 1. Introduction

In current studies, the production of nanoparticles has received a lot of interest as a simple way to generate plasmatic metal nanomaterials. Among the metal nanomaterials, gold nanoparticles (AuNps) are safe and effective in delivering pharmacological substances [1–4]. Hence, AuNps have been used as drug carriers, for plasmonic photothermal therapy (PPTT), cancer therapy, and DNA delivery [5]. The possibility of developing new materials for nanomedicine has gradually increased because of the great advances made in diagnosing and treating various diseases [6,7] based on the spherical AuNps preferentially bonded to cancer cells compared with binding to healthy cells [8]. Furthermore, stabilization of the size and shape of synthesized AuNps could offer different biological activities [9–11]. At the same time, the methods to synthesize nanoparticles face such challenges as the usage of toxic solvents, generation of hazardous byproducts, and excessive energy expenditure. There are three aspects of nanoparticle synthesis that should be considered to synthesize AuNps in an environmentally friendly way: selection of the solvent, reducing and stabilizing agents. Our study presents a rapid and green synthesis of AuNps using an aqueous extract of

*Suaeda japonica* as both a reducing agent and a stabilizing agent. The reduction process is performed in distilled water without other chemical agents. Green synthesis methods based on plants [12], fungi [13], and bacteria [14] are more beneficial as the usage of a harmless solvent, nontoxic reducing agents, and eco-friendly materials reduces the risks of biomedical applications [15]. Meanwhile, according to the basic principle for safety, biomedical agents should be effectively cleared from the body and have little accumulation within the organs. So far, blood circulation and organ clearance have played an important role in detoxifying nanoparticles from the human body [16].

Inflammation is a complex biological property of immunoreactions of vascular tissues responding to harmful stimuli. Inflammatory reactions are also associated with cancer, neurological disorders, inflammatory bowel disease, atherosclerosis, and coronary arteries [11,17–19]. Macrophages play an essential role in the human immune system by engulfing infectious agents and releasing proinflammatory mediators [20–22]. Several inflammatory illnesses are linked to proinflammatory mediators such as COX-2, iNOS, NO, tumor necrosis factor alpha (TNF-$\alpha$), and PGE$_2$ [23–25]. Thus, the levels of these proinflammatory mediators show significant meaning in the development of efficient inflammation treatment. Our earlier investigation found that gold nanoparticles produced from a medicinal plant extract showed anti-inflammatory properties in RAW 264.7 macrophages [26].

In Korea and Japan, *S. japonica* is a halophytic herb used as a crop commodity on tidal flats and salt marshes. The leaves of *S. japonica* change their color from green to red with the accumulation of betacyanin. In Korea, *S. japonica* has been used as an oriental medicinal plant to alleviate fever [27,28]. In recent studies, *S. japonica* showed strong antioxidant properties [26]. Hence, *S. japonica* was chosen as an effective medicinal herb to investigate its anti-inflammatory effects in a murine macrophage cell line. The gold nanoparticles biosynthesized by *S. japonica* were characterized by UV–Vis spectroscopy, DLS, EDX, FE-TEM, SAED, and XRD. Moreover, cytotoxicity was tested against the RAW 264.7 (murine macrophages), HaCaT (human keratinocyte cell line), and 3T3-L1 (murine pre-adipocytes) cells. Further, we investigated the anti-inflammatory potential of Sj-AuNps by inhibiting proinflammatory cytokines release and suppressing *iNOS*, *COX-2*, and *TNF-α* gene expression using dexamethasone (DEX) as a reference drug.

## 2. Materials and Methods

### 2.1. Materials

The leaves of *Suaeda japonica* were obtained from the Ginseng Bank, Kyung Hee University, Korea. Gold (III) chloride trihydrate (HAuCl$_4$·3H$_2$O) was purchased from Sigma-Aldrich Chemicals (St Louis, MO, USA). Other chemical materials used in this study were of analytical grade.

### 2.2. Preparation of an Aqueous Suaeda japonica Leaf Extract

As much as 10 g of *S. japonica* leaves were thoroughly grounded and extracted in 100 mL (1:10 ratio) of distilled water at 80 °C for 1 h. The *S. japonica* extract was filtered to remove solid particles. The filtered *S. japonica* aqueous leaf extract was stored at 4 °C for further use.

### 2.3. Green Synthesis of Gold Nanoparticles Utilizing the Suaeda japonica Aqueous Extract

Sterile distilled water was used to dilute the *S. japonica* aqueous extract to reach a concentration of 20%. The HAuCl$_4$·3H$_2$O solution was added into a prewarmed extract until reaching the final concentration of 1 mM. The mixture was heated at 80 °C using an oil bath. A steady color shift was observed, confirming the presence of nanoparticles. Following the synthesis, nanoparticles were collected by centrifugation at 15,000 rpm for 15 min at 4 °C. Repetitive washing with distilled water followed it. The nanoparticles were air-dried overnight.

### 2.4. Characterization of Sj-AuNps

The synthesis of AuNps may be confirmed by measuring the absorbance spectra of aliquots of the reaction mixture with a UV–Vis spectrophotometer (Ultrospec 2100 Pro, Amersham Biosciences, Freiburg, Germany) in the 300–800 nm range; the *S. japonica* aqueous extract is considered as a reference. DLS for nanoparticle size analysis was acquired at 25 °C utilizing an ELSZ-2000 series particle size analyzer (Otsuka Electronics Photal, Osaka, Japan). Pure water was utilized as a dispersive medium (refractive index of 1.3328, viscosity of 0.8878 cP, dielectric constant of 78.3). The EDX, FE-TEM, SAED techniques were used to examine the morphology, distribution, and purity of synthesized AuNps, which were determined using a multi-functional, 200 kV-operated electron microscope (JEM-2100F, JEOL, Tokyo, Japan). The partly purified pellet solution droplets were placed onto a carbon-coated copper grid to create nanoparticle samples. It was put to FE-TEM after drying at 60 °C. D8 Advance was subjected to XRD examination (Bruker D8 Advance, Bruker AXS, Karlsruhe, Germany). The instrument was operated with Cu-K$\alpha$ radiation ($\lambda$ = 1.54 Å) at 40 kV and 40 mA. The samples were scanned at a rate of 6°/min with an interval of 0.02° throughout a range of 20–80°. The average crystallite diameter of gold nanoparticles was determined using the Debye–Scherrer equation: D = 0.9 $\lambda$/$\beta$ cos $\theta$, where D denotes the crystallite size (nm), the wavelength of Cu-K$\alpha$ radiation (nm), $\beta$—the full width at half maximum (radians), and $\theta$—half of the Bragg angle (radians). These characterizations of Sj-AuNps procedures were based on previous research [29,30].

### 2.5. In Vitro Evaluation of the Cell Viability Assay

Cells 3T3-L1, HaCaT, and RAW 264.7 were cultivated at 37 °C in a humidified incubator with 5% $CO_2$. The MTT (3-(4,5-dimethylthiazol-2-yl)-2–5-diphenyltetrazolium bromide) test was used to assess the cell viability of Sj-AuNps. In a 96-well plate, $10^5$ cells mL$^{-1}$ were cultured (NEST, Brooklawn, NJ, USA). After 24 h, the cells were treated with various doses of Sj-AuNps for 48 h, followed by adding 20 μL of the MTT reagent (5 mg mL$^{-1}$) to each well, and incubated for 4 h. The supernatant was replaced with 100 μL of dimethyl sulfoxide (DMSO) and agitated for 30 min to dissolve the formazan crystals. Finally, the quantification of absorbance of each colored solution was performed using an enzyme-linked immunosorbent assay (ELISA) reader at 570 nm (tested wavelength) with a reference wavelength of 630 nm. Three independent replications were performed.

### 2.6. Measurement of NO, PGE$_2$, and TNF-$\alpha$ Production

The RAW 264.7 cells were cultured for 24 h with 1 g mL$^{-1}$ LPS and Sj-AuNps. The supernatant was then collected for the subsequent experiments. Culture supernatant (100 μL) was combined with an equivalent proportion of the Griess reagent. The outcome was determined using an ELISA reader set to 540 nm. NO, PGE$_2$, and TNF-$\alpha$ were quantified using ELISA kits following the manufacturer's procedure (R&D Systems, Minneapolis, MN, USA).

### 2.7. Gene Expression Studies

The RAW 264.7 macrophages were plated at a density of $1 \times 10^6$ cells mL$^{-1}$ in a six-well plate. After an overnight incubation period, 24 h of treatment with or without varying doses of Sj-AuNps in the presence or absence of LPS stimulation were added. Total RNA was extracted using the TRIzol reagent from Sigma-Aldrich Chemicals. The cDNA synthesis was carried out in accordance with the supplier's instructions (Thermo Scientific, Waltham, EU, Lithuania). Using the primers shown in Table 1, qRT-PCR was carried out. The relative gene expression levels were normalized to the amount of glyceraldehyde 3-phosphate dehydrogenase (GAPDH) expression. mRNA was determined by using the delta cycle threshold (Ct) method [31].

**Table 1.** The list of primer sequences.

| Primer Name | Sequence | Tm (°C) |
|:---:|:---:|:---:|
| iNOS | Forward: 5′-GTG GTG ACA ACG ACA TTT GG-3′ | 57.3 |
| | Reverse: 5′-GGC TGG ACT TTT CAC TCT GC-3′ | 59.3 |
| COX-2 | Forward: 5′-GGA TGC GCT GAA ACG TGG A-3′ | 58.8 |
| | Reverse: 5′-CAG GAA TGA GTA CAC GAA GCC-3′ | 59.8 |
| TNF-α | Forward: 5′-AGT CCG GGC AGG TCT ACT TT-3′ | 59.3 |
| | Reverse: 5′-GCA CCT CAG GGA AGA GTC TG-3′ | 61.4 |
| GAPDH | Forward: 5′-CAA GGT CAT CCA TGA CAA CTT TG-3′ | 59.4 |
| | Reverse: 5′-GTC CAC CAC CCT GTT GCT GTA G-3′ | 64.6 |

### 2.8. Immunofluorescence Staining

The RAW 264.7 cells were cultivated overnight on eight-well culture slides (SPL Life Sciences Co., Ltd., Pocheon, Korea). The cells were pretreated with Sj-AuNps for 2 h before being stimulated with LPS (1 $\mu$g mL$^{-1}$) for 2 h. The slides were carefully washed with phosphate buffer saline (PBS) before being fixed in 3.7% formaldehyde and permeabilized for 10 min with 0.5% Triton X-100. The slides were then treated overnight at 4 °C with rabbit monoclonal anti-NF-κB p65 antibodies (1:100 dilution, Santa Cruz Biotechnology, Santa Cruz, CA, USA). The slides were washed and incubated in the dark for 1 h with Alexa Fluor-488 labelled goat anti-rabbit IgG (1:200; Cell Signaling Technology, Beverly, MA, USA). For 10–15 min, the nuclei were stained with 40-6-diamidino-2-phenylindole (DAPI) (10 mg mL$^{-1}$; Sigma-Aldrich Co., St. Louis, MO, USA). The cells were photographed with an inverted research fluorescence microscope (Carl Zeiss, Axiovert 200M, Oberkochen, Germany) [32,33].

### 2.9. Statistical Analysis

The GraphPad 6.04 software was used for statistical analysis (San Diego, CA, USA). The results are shown as the mean SD. The statistical significance of differences in values between the treated and untreated groups was determined using two-way ANOVA and Student's *t*-test. At $p \leq 0.05$, differences in findings were considered significant.

## 3. Results and Discussion

The overall graphical representation of the *S. japonica*-mediated green synthesis of Sj-AuNps and its biological activities are illustrated in Figure 1.

### 3.1. Green Synthesis of Sj-AuNps

Standardization of the reaction conditions was achieved to regulate the optimal reaction temperature and time in this study. According to the results (Figure 2a,b), the reduction process in the yellowish reaction mixture was confirmed by a color shift to a deep purple. As much as 1 mM HAuCl$_4$·3H$_2$O was added to the 20% aqueous *S. japonica* extract for 1.5 min at 80 °C as the optimum factor in the green synthesis of Sj-AuNps. A broad peak that started to grow at around 400 nm was similar to the absorption spectrum of the Au (III)–CTAB complex solution as previously reported for Au nanorods [34], which indicated the appearance of the Au (III) ions band. In addition, as incubation temperature and time changed, the peaks became increasingly less prominent and broadened, demonstrating the appearance of agglomeration and instability of nanoparticles [35].

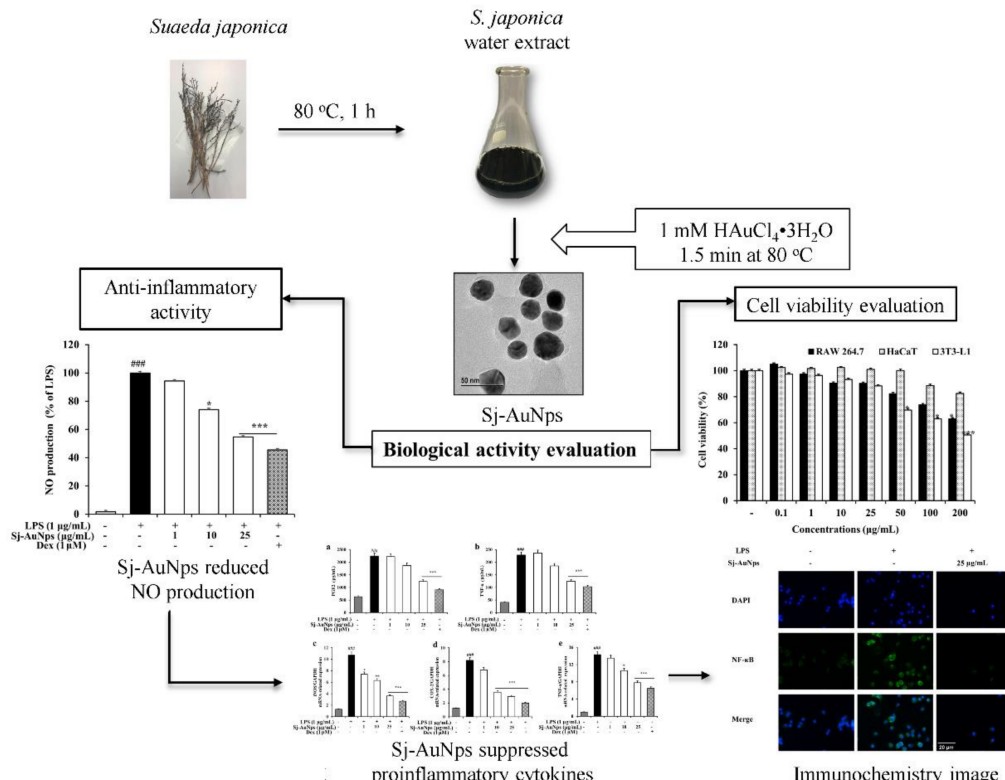

**Figure 1.** The overall graphical representation of the *Suaeda japonica*-mediated green synthesis of gold nanoparticles (Sj-AuNps) and its biological effects. *: $p \leq 0.05$; **: $p \leq 0.01$; ***: $p \leq 0.001$; ### (only LPS treated): $p \leq 0.01$.

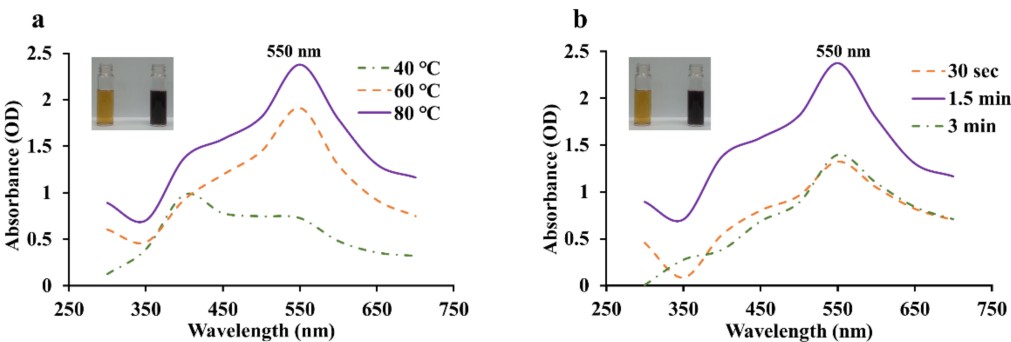

**Figure 2.** Temperature- (**a**) and time-dependent (**b**) UV–Vis spectrometry demonstrates the formation of Sj-AuNps.

### 3.2. Characterizations of Sj-AuNps

Following the reduction of $Au^{3+}$ to $Au^0$, UV–Vis spectroscopy revealed the most significant peak at 550 nm. The SPR phenomenon is aided by the unique absorbance peak [12,36]. FE-TEM was used to identify the morphology of Sj-AuNps. FE-TEM analysis indicated that the Sj-AuNps were virtually spherical, with diameters ranging from 20 to 30 nm (Figure 3a–c). EXD spectroscopy was used to determine the quality of the biosynthesized gold nanoparticles. The nanoparticles' EDX spectrum revealed the strongest optical absorbance band peak at 2.3 keV, which matches the typical peak of nanosized metallic gold (Figure 3f). The Sj-AuNps elemental mapping findings revealed the distribution of gold elements in the FE-TEM picture of the partly purified gold nanoparticles. Figure 3f depicts the distribution of gold in the FE-TEM picture. The electron picture obviously shows the distribution of elemental gold as a major element in the nanoparticles. The elemental mapping results validated the gold nanoparticles' spherical form even further.

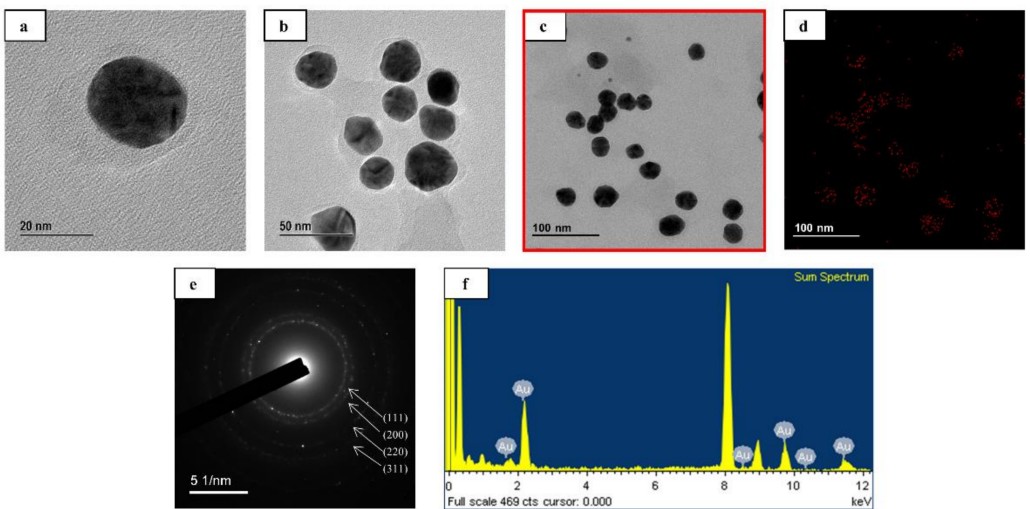

**Figure 3.** FE-TEM (**a**–**c**), elemental mapping (**d**), SAED (**e**), and EDX (**f**) expose the morphology and chemical properties of Sj-AuNps.

XRD measurement revealed the crystalline character of the biologically produced gold nanoparticles. Figure 4 depicts the intense diffraction patterns of biosynthesized gold nanoparticles in the spectrum of 2θ values spanning from 20 to 80°. The intensities recorded at the three lattice planes (i.e., (200), (220), and (311)) were much lower than the intensities obtained at the 111 planes, implying that the nanoparticles are largely constituted of (111) orientations. The average crystallite size of gold nanoparticles was determined using the Scherrer equation. The gold nanoparticles produced had an average crystallite size of 8.75 nm. The (111) plane of the nanoparticles is also confirmed by the SAED pattern (Figure 3e).

Based on Bragg's reflection, the XRD and SAED data show that the biosynthesized gold nanoparticles are fundamentally crystalline in nature and face-centered cubic.

The size distribution profiles of biosynthesized gold nanoparticles were obtained using the DLS method with respect to intensity, quantity, and volume. Figure 5 indicated that the gold nanoparticles' Z-average diameter was 268.0 nm with a polydispersity index (PDI) of 0.128. The difference in the average size of gold nanoparticles was investigated using FE-TEM and DLS since FE-TEM predicts the particle size of the nanoparticles. On the other hand, DLS measures the hydrodynamic diameter of the nanoparticles [37].

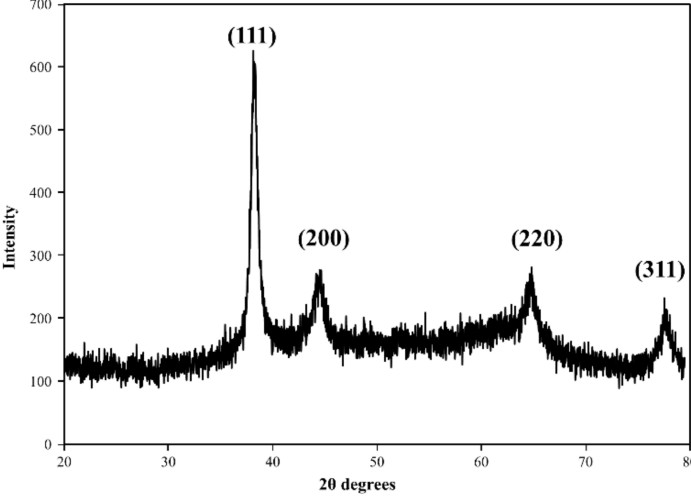

**Figure 4.** XRD analysis confirmed the crystallinity of the gold nanoparticles.

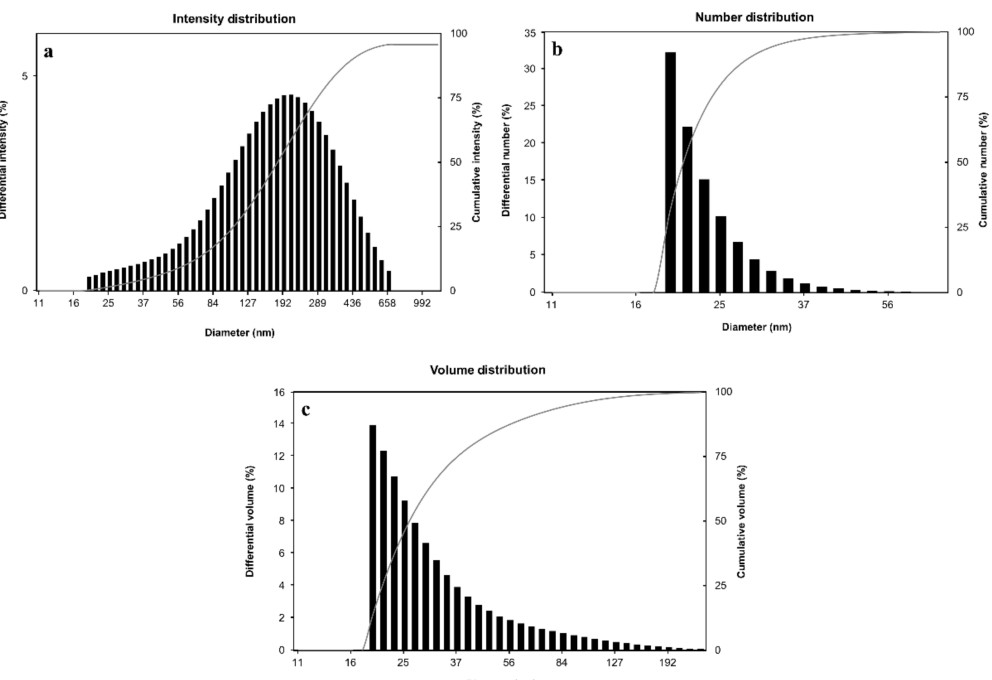

**Figure 5.** Sj-AuNps size distributions with respect to intensity (**a**), number (**b**) and volume (**c**).

### 3.3. In Vitro Applications of Sj-AuNps

One of the most common responses to injured tissues in our body is inflammation, which can be related to cancer, rheumatoid arthritis, and autoimmune disorders [38]. In vitro cytotoxicity of Sj-AuNps was determined against the RAW 264.7, HaCaT, and 3T3-L1 cells. As shown in Figure 6a, up to 25 $\mu$g mL$^{-1}$ of Sj-AuNps treatments exhibited non-cytotoxicity in these three cell lines. Thus, further experiments were performed using different concentrations (1, 10, and 25 $\mu$g mL$^{-1}$) of Sj-AuNps to determine the potential anti-inflammatory effects.

Macrophages are among the immune defense mechanisms in the human body that are capable of phagocytosis. They are also involved in the inflammatory response in terms of producing both iNOS and COX-2 through NF-κB activation [39]. Previous studies reported that NO production by iNOS and PGE$_2$ was derived from COX-2 and played an essential role during the inflammatory reaction process [40]. Therefore, to determine the anti-inflammatory effect of Sj-AuNps, the levels of NO and PGE$_2$ were measured. In our study, we also subjected determination of the NO and PGE$_2$ production in the LPS-induced RAW 264.7 cells, with or without Sj-AuNps treatment, to DEX as a positive control. In Figure 6b, our result indicated increasing nitrite levels in the LPS-induced RAW 264.7 cells compared with the basal level without LPS (1 $\mu$g mL$^{-1}$) treatment. The dose-dependent treatment of Sj-AuNps significantly decreased the NO production level. The qRT-PCR analysis in Figure 7c exhibited that the expression of *iNOS* at the gene level was also suppressed. Figure 7a shows that the LPS-induced PGE$_2$ production release was decreased by treating Sj-AuNps in a dose-dependent manner. Besides, the gene expression of *COX-2* significantly decreased with the Sj-AuNps treatment of the LPS-stimulated murine macrophages (Figure 7d). As a result, Sj-AuNps blocked the activities of both *iNOS* and *COX-2* at the mRNA level in the LPS-induced RAW 264.7 cells. In a previous study, the *S. japonica* extract suppressed PGE$_2$ production in LPS-stimulated RAW 264.7 cells at 200 $\mu$g mL$^{-1}$ [41]. Sj-AuNps could significantly inhibit the PGE$_2$ levels in 25 $\mu$g mL$^{-1}$ (Figure 7a).

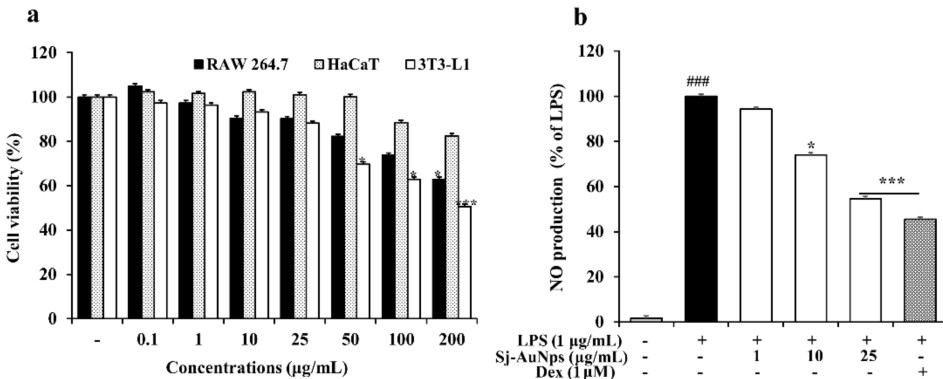

**Figure 6.** Evaluation of cell viability in the RAW 264.7, HaCaT, and 3T3-L1 cells 24 h after the Sj-AuNps treatment (**a**) and capability of Sj-AuNps to inhibit NO production in LPS-induced RAW 264.7 cells (**b**). *: $p \leq 0.05$; ***: $p \leq 0.001$; ### (only LPS treated): $p \leq 0.01$.

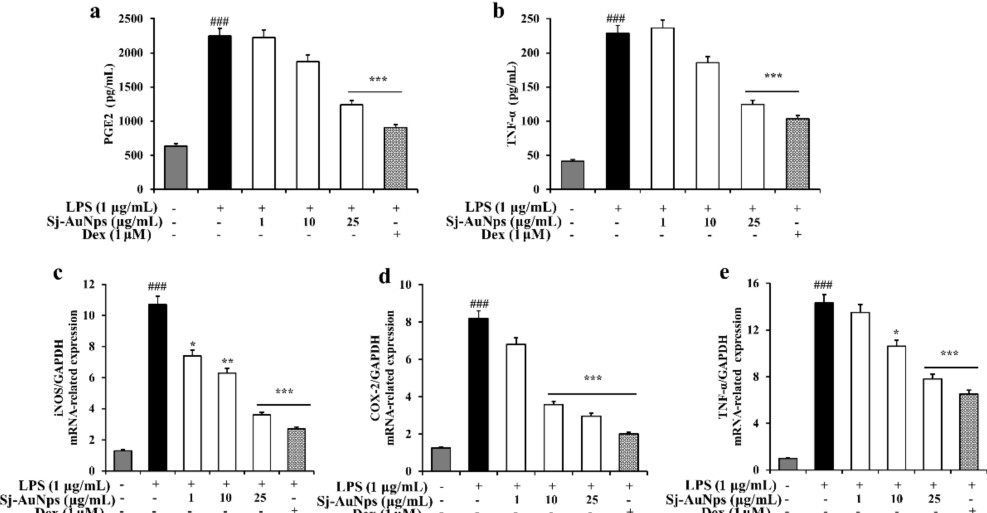

**Figure 7.** Determination of the reduction of the PGE$_2$ (**a**) and TNF-$\alpha$ (**b**) release. Expression of *iNOS* (**c**), *COX-2* (**d**), and *TNF-$\alpha$* (**e**) at the gene level. ### (only LPS treated): $p \leq 0.01$ compare with the negative control; *: $p \leq 0.05$; **: $p \leq 0.01$; ***: $p \leq 0.001$ compare with the negative control.

Activated macrophages and T cells produce TNF-$\alpha$ and other proinflammatory cytokines as part of the immune response [42,43]. Through ELISA testing of TNF-$\alpha$ production in LPS-induced murine macrophages, we evaluated the efficacy of Sj-AuNps to decrease proinflammatory mediators. Sj-AuNps reduced the LPS-induced TNF-$\alpha$ release in a dose-dependent manner, as demonstrated in Figure 7b. Following that, we used qRT-PCR to look at the *TNF-$\alpha$* gene level. TNF-$\alpha$ levels in the stimulated RAW 264.7 cells were reduced by Sj-AuNps (Figure 7e). TNF-$\alpha$ was reduced in our study, indicating that Sj-AuNps may have anti-inflammatory characteristics.

According to the immunofluorescence staining in Figure 8, the NF-κB density (green fluorescence) was distinctively enhanced at 2 h of LPS exposure, but LPS-induced nuclear translocation was significantly inhibited by Sj-AuNps cotreatment. The quantities of nuclear NF-κB p50 were increased after LPS exposure. Sj-AuNps significantly inhibited this LPS-induced nuclear translocation. Green synthesis nanotechnology provides a successful option for synthesizing new gold nanoparticles and suggests innovative new solutions for diseases [44,45]. Furthermore, the anti-inflammatory effects of Sj-AuNps are related to the suppression of the *iNOS* and *COX-2* genes during the treatment in the form of reduction of the production of nitrites and PGE$_2$. Meanwhile, the proinflammatory mediator TNF-$\alpha$ is suppressed at the gene level; Sj-AuNps can also inhibit the release of TNF-$\alpha$ as well. Anti-inflammatory progression suppressed nuclear-translocated NF-κB by Sj-AuNps.

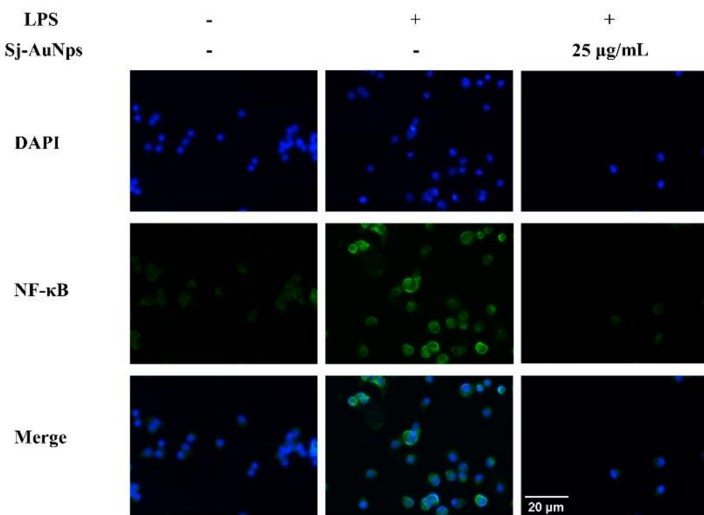

**Figure 8.** Immunofluorescence staining of NF-κB expression in the RAW 264.7 macrophages.

## 4. Conclusions

We chose the green synthesis approach and utilized naturally solidifying techniques fit for a broad scope of chances for an additional examination of nanoparticles in the biological field. Our study utilized an aqueous *S. japonica* leaf extract to synthesize gold nanoparticles without additional toxic chemicals. Sj-AuNps was acquired by biosynthesis and had a spherical shape (20–30 nm). The green-synthesized nanoparticles were analyzed in vitro for cytotoxic activities against murine preadipocytes (3T3-L1), murine macrophages (RAW 264.7), and a human keratinocyte cell line (HaCaT). Furthermore, we clarified the anti-inflammatory properties of Sj-AuNps through decreased proinflammatory cytokines and inflammatory mediator production. As a result, green synthesis of nanoparticles will be a potential therapeutic and pharmacological candidate.

**Author Contributions:** G.-Y.K. and Y.H. conceptualized and provided resources for the research. G.-Y.K., Y.H., S.B., B.-M.K., D.-C.Y. and S.-C.K. carried out the main experiments. All the six authors contributed to writing the manuscript. B.-M.K., S.-C.K. and J.S. edited and reviewed the manuscript. J.S. helped with the supervision of the work and provided funding. All authors have read and agreed to the published version of the manuscript.

**Funding:** This research was funded by a grant from the Ministry of Research and the Ministry of Education, Culture, Research, and Technology, Indonesia.

**Institutional Review Board Statement:** Not applicable.

**Informed Consent Statement:** Not applicable.

**Data Availability Statement:** Not applicable.

**Conflicts of Interest:** The authors declare no conflict of interest.

## Abbreviations

3T3-L1 = murine preadipocytes, AuNps = gold nanoparticles, Ct = delta cycle threshold, COX-2 = cyclooxygenase-2, CTAB = cetyltrimethylammonium bromide, DAPI = 40-6-diamidino-2-phenylindole, DEX = dexamethasone, DLS = dynamic light scattering, DMSO = dimethyl sulfoxide, EDX = energy-dispersive X-ray spectroscopy, ELISA = enzyme-linked immunosorbent assay, FE-TEM = field-emission transmission electron microscopy, GADPH = glyceraldehyde 3-phosphate dehydrogenase, HaCaT = human keratinocyte cell line, iNOS = inducible nitric oxide synthase, MTT = 3-(4,5-dimethylthiazol-2-yl)-2–5-diphenyltetrazolium bromide, NO = nitric oxide, PDI = polydispersity index, $PGE_2$ = prostaglandin E2, PBS = phosphate-buffered saline, PPTT = plasmonic photothermal, qRT-PCR = quantitative reverse transcription polymerase chain reaction, RAW 264.7 = murine macrophages, SAED = selected area electron diffraction, Sj-AuNps = *S. japonica* gold nanoparticles,

TNF-α = tumor necrosis factor alpha, UV–Vis = UV–visible, XRD = X-ray diffraction.

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
