# Peer review of "Gold Nanoparticles Green-Synthesized by the Suaeda japonica Leaf Extract and Screening of Anti-Inflammatory Activities on RAW 267.4 Macrophages"

_coatings, doi:10.3390/coatings12040460_

Round 1
Reviewer 1 Report
- The mechanism of the prepared nanoparticles in the inflammation should be discussed with the help of a schematic diagram.
- The authors should highlight how their nanoparticles are characteristically different or better, or comparable to other nanoparticles in the discussion.
- The introduction is too short. Nowadays, nanotechnology has extensive application in medicine. Generally, the authors should first discuss the application of nanotechnology in medicine.
- Are gold nanoparticles effective only in treating inflammation?!!! The authors discuss the anti-inflammatory effect of AuNPs, while AuNPs are performance tools in different aspects such as cancer therapy, drug delivery systems, and cosmetics. The authors should consider other aspects of the application of AuNPs in medicine. In this regard, the authors should be discussed and cite proper and related ref such as DOI: 10.3390/nano11113002.
- Nowadays, medicinal plants-based nanomaterials such as AuNPs have the main portion in green medicine. The author did not discuss recent ref in the biosynthesis of AuNPs using medicinal plants. In this context, the authors should be cited the most recent ref such as DOI: 10.3390/nano11082033.
- Detoxifying nanoparticles from a patient's body is a critical issue in nano-based therapy. It is recommended to explain the detoxification and filtration of idle AuNPs from the blood of the treated cancer patients in the discussion.
- The economic cost of AuNPs is one of the critical factors for their application in therapy goals. It is recommended that the authors pay attention to this issue on the right site.
- It is recommended that authors add a figure in visual confirmation of biosynthesized AuNPs
- The authors must have a logical reason for the size differences of nanoparticles measured by FE-TEM and DLS.
- “ S. japonica” Should be italic (in the Introduction)
Author Response
Thank you for all valuable suggestions, please see the attachment.

Reviewer 2 Report
Coating 1596553
Green synthesized gold nanoparticles by Suaeda japonica leaf extract and screening of anti-inflammatory activities on RAW 267.4 macrophage
In this paper the biosynthesis of gold nanoparticles from medicinal plants has been carried out. This is a modern strategy in biomedical research based on their exclusive properties, less toxicity and biocompatibility. The process is validated by UV-vis spectroscopy and other techniques to illustrate the nano-scale characterization. The green synthesized nanoparticles were analysed in vitro for cytotoxic activities against pre-adipocytes, macrophages, and human cell lines. They also clarified anti-inflammatory properties giving as a result, that these nanoparticles are a potential therapeutic and pharmacological candidates.
The paper has good quality and deserves to be published, but several details need to be previously improved:
The paper needs to take care of the acronyms and graphs.
I recommend including a short list of acronyms or to define each before cite it.
figures 1. It is necessary to increase the axis label size and symbols inside plots (a) and (b).
Figure 1b. The evolution with time seems rare. Please, what is the rational which explain the variation? Alternatively, is it correct the labelling of the UV-vis spectra with time?
What is assigned the broad peak around 400 nm? Have you tried some blank as reference? Nothing is said in the text.
Figures 4, 5 and 6. The quality of the figures needs to be improved: numbering, labelling, etc. In the present version the quality is no appropriate since it is difficult to appreciate most of details.
Author Response
Thank you for all valuable suggestions,
Reviewer 2
Green synthesized gold nanoparticles by Suaeda japonica leaf extract and screening of anti-inflammatory activities on RAW 267.4 macrophage
In this paper the biosynthesis of gold nanoparticles from medicinal plants has been carried out. This is a modern strategy in biomedical research based on their exclusive properties, less toxicity and biocompatibility. The process is validated by UV-vis spectroscopy and other techniques to illustrate the nano-scale characterization. The green synthesized nanoparticles were analysed in vitro for cytotoxic activities against pre-adipocytes, macrophages, and human cell lines. They also clarified anti-inflammatory properties giving as a result, that these nanoparticles are a potential therapeutic and pharmacological candidate.
The paper has good quality and deserves to be published, but several details need to be previously improved:
Comment 1: The paper needs to take care of the acronyms and graphs. I recommend including a short list of acronyms or to define each before cite it.
Response: Thank you for your valuable suggestion, we propose the abbreviation list which elaborate each acronym used in this manuscript, right before the introduction part. We also checked carefully about the acronyms in the manuscript and corrected them as required.
Comment 2: figures 1. It is necessary to increase the axis label size and symbols inside plots (a) and (b).
Response: Thank you for your valuable suggestion. We have modified the axis label size and symbols. We also add graphical representation of the green synthesis gold nanoparticle (Sj-AuNps) by Suaeda japonica and its biological activities as Figure 1. Due to this modification, we also changed the numbering for the following figures.
Comment 3: Figure 1b. The evolution with time seems rare. Please, what is the rational which explain the variation? Alternatively, is it correct the labelling of the UV-vis spectra with time?
Response: Thank you for your valuable suggestion. We aimed to optimize the reaction time and temperature for the bio-reduction of AuNps by using UV-Vis spectral analysis. And we have re-arranged the manuscript to explain the synthesis process.
The standardization of reaction conditions was achieved to regulate the optimal reaction temperature and time in this study. According to the results (Figure 2a and 2b), the reduction process occurring to the yellowish reaction mixture was confirmed by a color shift into deep purple. 1mM HAuCl4•3H2O as the final concentration was added to the 20% S. japonica aqueous extract for 1.5 min at 80 ℃ as the optimum factor in the process of green synthesizing Sj-AuNPS. As mentioned previously, when a 20% aqueous extract was used to make gold nanoparticles, a color shift in the reaction mixture was noticed. The aqueous extract diluent was yellowish before being warmed in the 80°C of the oil bath. Afterward, 1 mM of gold salt was added to the diluent, the color of the reaction mixture consequently shifted to deep purple (Figure 1a and 1b).
In addition, as incubation temperature and time were changed, the peaks became increasingly less prominent and broadened, demonstrating the appearance of agglomeration and instability of nanoparticles [35].
We also added the reference,
- Markus, J.; Wang, D.; Kim, Y.-J.; Ahn, S.; Mathiyalagan, R.; Wang, C.; Yang, D. C. Biosynthesis, characterization, and bioactivities evaluation of silver and gold nanoparticles mediated by the roots of Chinese herbal Angelica pubescens Maxim. Nanoscale Res. Lett.2017, 12, 1-12.
Comment 4: What is assigned the broad peak around 400 nm? Have you tried some blank as reference? Nothing is said in the text.
Response: Thank you for your valuable comment. We have used the S. japonica aqueous extract as the reference (in 2.4. Characterization of Sj-AuNps). We confirmed the broad peak started to grow at around 400 nm is similarly to the absorption spectrum of Au (III)-CTAB complex solution as previously reported for Au nanorods [34], which indicated the appearance of the Au (III) ions band.
We also added the reference,
- Attia, Y. A.; Buceta, D.; Blanco-Varela, C.; Mohamed, M. B.; Barone, G.; López-Quintela, M. A. Structure-directing and high-efficiency photocatalytic hydrogen production by Ag clusters. J. Am. Chem. Soc. 2014, 136, 1182-1185.
Comment 5: Figures 4, 5 and 6. The quality of the figures needs to be improved: numbering, labelling, etc. In the present version the quality is no appropriate since it is difficult to appreciate most of details.
Response: Thank you for your valuable suggestion. We have edited the figures quality in the revised manuscript to make it more readable.
Round 2
Reviewer 1 Report
It is acceptable now.
